# Direct transport vs secondary transfer to level I trauma centers in a French exclusive trauma system: Impact on mortality and determinants of triage on road-traffic victims

Sophie Rym Hamada[1,2]*, Nathalie Delhaye[3], Samuel Degoul[4], Tobias Gauss[5], Mathieu Raux[6], Marie-Laure Devaud[7], Johan Amani[8], Fabrice Cook[9], Camille Hego[10], Jacques Duranteau[11☯], Alexandra Rouquette[12,13☯], the Traumabase Group[¶]

1 Université paris Sud, Department of Anesthesiology and Critical Care, AP-HP, Bicêtre Hôpitaux Universitaires Paris Sud, Le Kremlin Bicêtre, France, 2 CESP, INSERM, Université paris Sud, UVSQ, Université Paris-Saclay, Paris; CESP, INSERM, Maison de Solenn, Paris, France, 3 Sorbonne Université and Department of Anesthesiology and Critical Care, AP-HP, Hôpitaux Universitaires Pitié-Salpêtrière, Paris, France, 4 Groupe Hospitalier de la Région de Mulhouse et Sud-Alsace, Department of Anesthesiology and Surgical Intensive Care, Mulhouse, France, 5 Hôpitaux Universitaires Paris Nord Val de Seine, Department of Anesthesiology and Critical Care, AP-HP, Hôpital Beaujon, Clichy, France, 6 Sorbonne Université, INSERM, UMRS1158 Neurophysiologie Respiratoire Expérimentale et Clinique; AP-HP, Groupe Hospitalier Pitié-Salpêtrière Charles Foix, Département d'Anesthésie Réanimation, Paris, France, 7 SAMU 95, Centre Hospitalier René Dubos, Pontoise, France, 8 SAMU 78, Centre Hospitalier de Versailles, Le Chesnay, France, 9 Université Paris Est, Department of Anesthesiology and Critical Care, APHP, Hôpital Henri Mondor, Créteil, France, 10 Hôpitaux Universitaires Paris Nord Val de Seine, Department of Anesthesiology and Critical Care, AP-HP, Beaujon, Clichy, France, 11 Université Paris Sud, Department of Anesthesiology and Critical Care, AP-HP, Bicêtre Hôpitaux Universitaires Paris Sud, Le Kremlin Bicêtre, France, 12 CESP, INSERM, Univ. Paris-Sud, UVSQ, Université Paris-Saclay, Paris, France (Postal address: CESP, INSERM, Maison de Solenn, Paris, France, 13 Bicêtre Hôpitaux Universitaires Paris Sud, Public Health and Epidemiology Department, APHP, Le Kremlin-Bicêtre, France

☯ These authors contributed equally to this work.
¶ Traumabase Group: the complete membership of the author group can be found in the Acknowledgments
* sophiehamada@hotmail.com

**Data Availability Statement:** Data are from the Traumabase, a collaborative project that consists of trauma practitioners, from different centres all

## Abstract

### Background

Transporting a severely injured patient directly to a trauma center (TC) is consensually considered optimal. Nevertheless, disagreement persists regarding the association between secondary transfer status and outcome. The aim of the study was to compare adjusted mortality between road traffic trauma patients directly or secondarily transported to a level 1 trauma center (TC) in an exclusive French trauma system with a physician staffed prehospital emergency medical system (EMS).

### Methods

A retrospective cohort study was performed using 2015–2017 data from a regional trauma registry (Traumabase®), an administrative database on road-traffic accidents and prehospital-EMS records.

over the country, sharing the registry for research and public health issues. Even though datasets are de-identified, the National Commission for data protection (CNIL) has imposed restrictions on data sharing since they contain sensitive information on trauma. In the present study, the Traumabase data have been linked to data owned by the Ministry of the Interior. Restrictions to the sharing of these data are ruled by the body of the Road Safety Office in the Ministry of the Interior and a special convention has been signed to secure the data. For data access requests, interested researchers can contact the Traumabase scientific committee with the following email address: contact@traumabase. eu (secretary of the scientific committee- Dr. Anatole Harrois).

**Funding:** This research was funded by a grant from the National Inter-ministerial Observatory of Road Safety (ONISR) belonging to The Ministry of the Interior. The French Regional health agency has funded a part of the trauma registry. The funders had no role in study design, data collection and analysis, decision to publish, or preparation of the manuscript.

**Competing interests:** The authors have declared that no competing interests exist.

**Abbreviations:** AC/AP therapy, anticoagulant and/ or antiplatelet therapy; AIS, abbreviated injury scale; ASA, American society of anesthesiologists; AUC, area under curve; BMI, body mass index; CI, confidence interval; CNS, central nervous system; EMS, emergency medical service; GCS, Glasgow coma scale; HR, heart rate; HS, hemorrhagic shock; ICU, intensive care unit; ISS, injury severity score; LOS, length of stay; Max, maximal; Min, minimal; MOF, multiple organ failure; MVA, motor vehicle accident; OR, odds ratio; RAAR, road accident analysis report; RTA, road traffic accident; SAMU, Service d'Aide Médicale Urgente; SAP, systolic arterial blood pressure; SAPS II, simplified acute physiologic score II; SMUR, Service Mobile d'Urgence et de Réanimation; SOFA, sequential organ failure assessment; SpO$_2$, peripheral oxygen saturation; ST, secondary transfer; TC, trauma center; TCA, traumatic cardiac arrest; TRISS, trauma related injury severity score.

Multivariate logistic regression models were computed to determine the role of the modality of admission on mortality and to identify factors associated with secondary transfer. The primary outcome was day-30 mortality.

Results: During the study period, 121.955 victims of road-traffic accident were recorded among which 4412 trauma patients were admitted in the level 1 regional TCs, 4031 directly and 381 secondarily transferred from lower levels facilities. No significant association between all-cause 30-day mortality and the type of transport was observed (Odds ratio 0.80, 95% confidence interval (CI) [0.3–1.9]) when adjusted for potential confounders. Patients secondarily transferred were older, with low-energy mechanism and presented higher head and abdominal injury scores. Among all 947 death, 43 (4.5%) occurred in lower-level facilities. The population-based undertriage leading to death was 0.15%, 95%CI [0.12–0.19].

## Conclusion

In an exclusive trauma system with physician staffed prehospital care, road-traffic victims secondarily transferred to a TC do not have an increased mortality when compared to directly transported patients.

## Background

Major trauma and road traffic accidents in particular remain an important cause of death and disability worldwide, especially among young and economically active adults [1]. Part of the answer is the organization of trauma care into structured regional trauma systems embedded into guidelines and along a national policy. Efforts and experiences from the United States [2,3], United Kingdom [4,5], Germany [6], Norway [7], or Netherlands revealed improved outcomes over the last 20 years. The provision of structured and coordinated trauma care from the scene to the rehabilitation unit has a far higher impact on outcome than any single medical intervention [2].

An *exclusive* trauma system is based on the provision of care by a limited number of highly specialized designated hospitals, so-called level 1 trauma centers (TC). Providing trauma care within a TC is now known to save lives and prevent long-term disability [2], so the direct transportation of severely injured patients to designated centers, while bypassing closer non-specialized facilities, is considered optimal. Nevertheless, few studies have analyzed the relationship between mortality and direct or secondary transport to TC and they generated conflicting results [8–12]. Those studies were exclusively conducted in paramedic staffed prehospital emergency medical systems without physician-staffed pre-hospital teams [13,14]. In France, as in some other European countries, the organization of prehospital care and trauma systems is systematically provided and coordinated by specialized physicians [15].

The main objective of this study was to compare adjusted intrahospital mortality between road traffic patients directly transported to a referral level 1 TC and patients secondarily transferred from a lower level facility. Secondary goals were 1/ to identify factors associated with secondary transfer (considered as undertriage), and 2/ to evaluate the proportion of patients deceased in lower level facilities before admission to a level 1 TC (considered as undertriage) and calculate a population-based undertriage.

## Methods

### Study design and setting

This is a retrospective cohort study conducted on prospectively collected data in the administrative area of Paris (Île de France region) which is organized as an exclusive trauma system with six level 1 TCs designated by the Regional Health Agency. The Île de France region gathers 12 million inhabitants, so about 20% of mainland France population and records an average of 42.000 road traffic accidents (RTA) each year. The emergency medical system (EMS) of the Île de France region is described in Fig 1, and also detailed in previous studies by Hamada *et al.* [16,17].

In France, dispatching physicians located in centralized SAMUs are available 24/7 and decide which emergency vector (paramedic-staffed ambulance or physician-staffed mobile intensive care unit) is to be deployed based on information provided during the emergency call. After clinical assessment, patient's triage is decided jointly by the physician on scene (when present) and the dispatching physician according to a national algorithm [18] and to the Mechanism, Glasgow coma scale and Arterial Pressure (MGAP) score [19]. All patients suspected of severe trauma are necessarily cared for by a physician-lead enhanced care team in a mobile intensive care unit and directly transported to a level 1 TC. The patient can also be transported to lower level of care facilities, with or without physician escort. Sometimes severe patients are first admitted to lower level of care facilities to allow initial stabilization, then transferred to a specialized center.

### Population and databases

This study included all the patients involved in a RTA in the Île de France region between January 1, 2015 and December 31, 2017.

All patients admitted to one of the six level 1 TCs of the region were prospectively registered in the regional trauma registry (Traumabase, http://traumabase.eu). The Traumabase obtained approval from the Advisory Committee for Information Processing in Health Research (CCTIRS, 11.305bis) and from the National Commission for data protection (CNIL, 911461), and meets the national institutional review board requirements (Comité de Protection des Personnes Paris VI, Paris France). The information is given to the patients (or next of kin, in case of comatose patients) about the existence of the trauma registry, but the institutional review board waived the need for informed consent. Data are anonymized from the time they are collected in the case report file. Sociodemographic, clinical, biological and therapeutic data (from the prehospital phase to the discharge of hospital) are systematically collected for all recorded patients.

In parallel, for all RTA involving any vehicle in France, the Police collects descriptive variables on the road user, vehicle, location and conditions of the crash. These road accident analysis reports (RAAR) are sent to the Ministry of the Interior once the 30-day survival status is completed. The RAARs constitute a nation-wide database of more than 10 million accidents since 1953.

Records from the RAAR and Traumabase were linked in order to evaluate the frequency of patients deceased in non-TC facilities (considered as inappropriate triage). The process of data management and linking is detailed in the supporting information (S1 File) and synthetized in the Flow chart (Fig 2). For individuals recorded as dead in the RAAR, but who could not be linked to any patient recorded in the Traumabase, archives from the 8 central dispatch centers of the Île de France region (SAMU) were examined to establish where the death occurred (either on scene or in an identified hospital).

## Variable definitions and outcome

The primary outcome was the 30-day in-hospital all-causes mortality of RTA patients alive on admission to the TC. The causes of death were determined by the physicians in charge. The modality of admission was defined as *direct transport* if the patient was directly transported from the scene to a level 1 TC, and as *secondary transfer* if the patient was transferred to a level 1 TC within the first 48 hours after the RTA but after a previous admission in a lower level facility.

Available prehospital variables included the mechanism of accident and distance to TC (categorized in inner or outer region as in Fig 1), demographic variables (age and gender) and worst vital signs during prehospital phase among: minimal Glasgow Coma Scale (GCS), maximal heart rate (HR), minimal systolic arterial blood pressure (SAP), minimal peripheral oxygen saturation (SpO$_2$) [20], early on scene capillary hemoglobin concentration [21], MGAP score [19], tracheal intubation and vasopressor administration. The assessment of high speed was reported according to the Vittel criteria and the total number Vittel criteria was quantified [18]. The Abbreviated Injury Scale (AIS) of body regions version 2005 and the Injury Severity Score (ISS) were used as surrogate measures of injury severity. Comorbidities were classified according American Society of Anesthesiologists physical status classification (ASA) and antiplatelet or anticoagulation therapy intake.

Available variable on admission were hemodynamic status, initial blood sample results, surgery within the first day and the Simplified Acute Physiologic Score II (SAPS2) and both

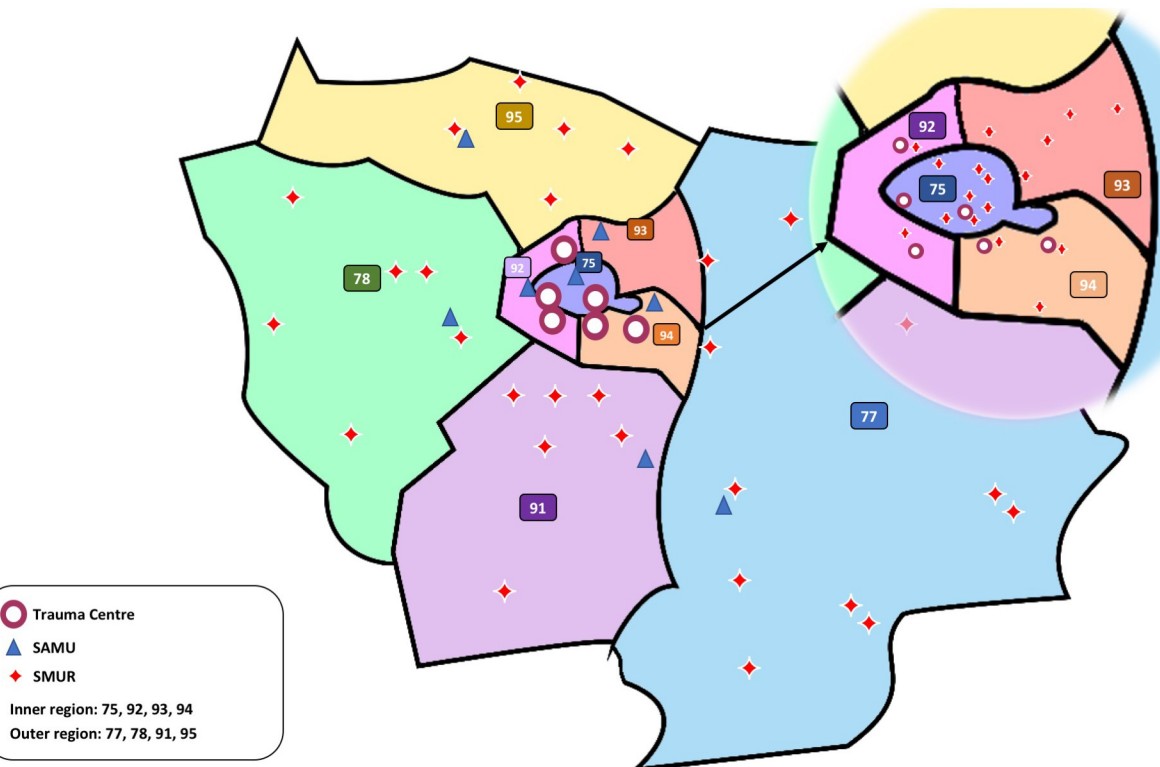

**Fig 1. Map of the region Île-de-France with trauma center, SAMU dispatch center and ambulance stations location.** The *Île de France* region gathers 8 central emergency medical system dispatch centers (SAMU- Service d'Aide Médicale Urgente), 43 prehospital enhanced mobile care team stations (SMUR- Service Mobile d'Urgence et de Réanimation) and 6 level 1 Trauma Centers (TCs). The region is organized as an exclusive trauma system, conceiving severe trauma care only in the 6 designated TC among all existing tertiary care facilities (n = 87).

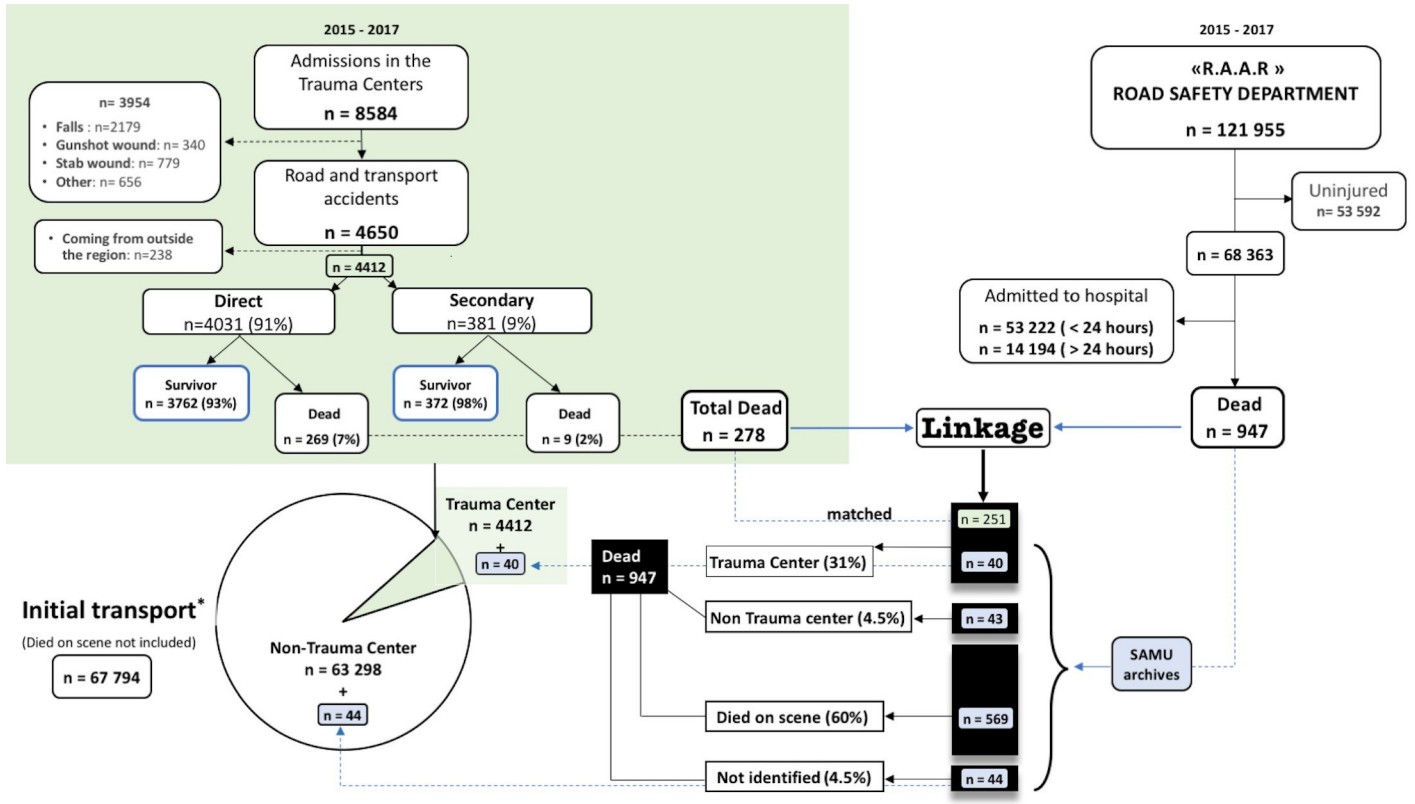

**Fig 2. Flow charts of the study.** Green filling: Traumabase data. Black boxes: RAAR and Traumabase linkage to identifiy the place of death of patients dead outside the trauma centre. Blue filling: Data extracted from SAMU archives. Pie: Population of hospital admissions extracted from RAAR, Traumabase and SAMU archives joint analysis

intensive care unit and hospital length of stay at discharge. The expected probability of survival was calculated using the Trauma Related Injury Severity Score (TRISS) with most recent coefficients [22,23] using a respiratory rate of 20 min$^{-1}$ in all patients [20].

Undertriage, defined as severe patients not ending in a TC, was computed according to three approaches: 1/ TC-based undertriage: (number of secondarily transferred patients) ÷ (number of admission in the TCs), 2/ Population-based undertriage leading to death according to the Cribari matrix method [24]: (number of death among patients first admitted in non TC) ÷ (number of admission in non TC) (Fig 2), 3/ Considering the boundaries of the 95% confidence interval of the observed mortality in patients directly admitted to the TC, we computed (with a Chi2 approach) the hypothetic systemic undertriage rate range outside which the crude mortality of inappropriately triaged patients (dead after initial transport in non-TC) would be significantly different than in TC-directly admitted patients.

Overtriage, defined as transport to the trauma center of patients ending with ISS < 16, was computed according to the Cribari method.

### Statistical analyses

The report follows the Strobe guidelines [25]. Continuous data were described as mean (standard deviation) or median [1$^{st}$– 3$^{rd}$ quartile] according to their distribution, and categorical variables as count (percentages) [26]. Bivariate analyses were performed using Chi$^2$ test and

Student t tests (or Mann-Whitney test) depending on variable type, to identify covariates associated with 30-day mortality and/or with secondary transfer.

Two multivariate logistic models were computed 1/ to identify the determinants of secondary transfer among prehospital available characteristics and 2 /to assess the association between secondary transfer and 30-day mortality adjusted on prehospital characteristics and characteristics available at admission. All confounders, i.e. the variables associated with the outcomes (p-value <0.05) in univariate analyses, were selected for multivariate analyses except in case of collinear variables (Spearman correlation coefficients >0.8), where the most clinically relevant variable was selected [27]. Initial physiologic variables (arterial pressure, heart rate and oxygen saturation) were withdrawn from the model because they were missing not at random in the secondary transfer group for more than 22% (vs <2% in the direct transport group). Multiple imputation via factor analysis for mixed data was used to handle missing data [28] as the average rate of missing data was 5.5% (maximum 15.7%) for variables candidate to multivariate analyses. Then, the variables were entered altogether into a multivariate logistic regression model and selection was performed using a backward stepwise procedure to optimize the Akaike criterion and the area under the receiver operating curve (AUC). The clinically relevant pairwise univariate interactions were tested. Model calibration was assessed using the Hosmer and Lemeshow statistic [29]. Model discrimination was assessed using the AUC and a bootstrap methodology (1000 samples) was used to quantify any optimism (averaged difference between the apparent AUC of the model developed on each bootstrap sample and its AUC on the original sample) in the final prediction model [30]. A sensitivity analysis was performed with a model on "complete cases" gathering all prehospital physiologic variables.

Based on an hypothesized prevalence of secondary transfer of 10% and an hypothesized overall mortality of 6%, our sample size was sufficient to show a mortality difference of 4% between the two groups with a power of 80% and a type one error of 5%[31]. All tests were two-sided and a $p \leq 0.05$ was considered significant. R 3.5.1 software (R Foundation for Statistical Computing, Vienna, Austria) was used for analyses.

## Results

During the study period, 121 955 road traffic accidents were recorded by the Police (RAAR) and 4412 patients were admitted to the regional TCs following a road traffic accident in the Île de France region (Traumabase). Direct transportation to TC occurred for 4031 patients (91%), whereas 381 (9%) patients were secondarily transferred (Flow chart Fig 2). Overall, 278 (6%) in-hospital deaths were recorded over the 30 days following admission in the TC. The causes of death are detailed in the Table 1. There was no significant difference in the causes of death between the direct and secondary transfer groups (p = 0.07).

The characteristics of patients directly and secondarily transferred to the TCs are presented in Table 2 (prehospital available characteristics) and Table 3 (severity and hospital course).

The multivariate analysis identified 10 prehospital variables independently associated with the type of transport (Table 4). Identified risk factors for secondary transfer were age (Odds ratio (OR) 1.4, 95% confidence interval (CI) [1.1–1.9] and 1.9, 95% CI [1.3–2.8] for ages 45 to 64 years and 65 to 100 years respectively), bicycle or pedestrian accidents (OR 2.2, 95% CI [1.4–3.4] and 1.5, 95% CI [1.0–2.1] respectively), high GCS score (OR 1.4, 95% CI [1.3–1.5]) and antiplatelet or anticoagulation therapy intake (OR 2.0, 95% CI [1.3–3.1]). In contrast, entrapment (OR 0.6, 95% CI [0.4–0.9]), high speed mechanisms (OR 0.6, 95% CI [0.5–0.8]) and pelvic injuries (OR 0.5, 95% CI [0.3–0.6]) were associated with a direct transfer. To notice, the only pairwise univariate interaction retained was between head and abdomen injury

**Table 1. Characteristics of patients deceased during the 30 days after admission depending on the cause of death.**

| | CNS | | HS | Hypoxia/ Anoxia | MOF | Septic shock | TCA | | Unknown | |
|---|---|---|---|---|---|---|---|---|---|---|
| | Direct | ST | Direct | Direct | Direct | Direct | Direct | ST | Direct | ST |
| n | 172 | 5 | 34 | 4 | 39 | 2 | 10 | 1 | 8 | 3 |
| **Demography and outcome** | | | | | | | | | | |
| Age (year) | 40 [26–60] | 82 [66–87] | 46 [33–64] | 64 [45–76] | 46 [29–65] | 72 [69, 74] | 43 [32–52] | 56 | 65 [51–70] | 85 [85–88] |
| Male (%) | 123 (72) | 3 (60) | 24 (71) | 3 (75) | 31 (80) | 1 (50) | 8 (80) | 0 | 7 (88) | 2 (67) |
| ICU LOS | 3 [2–6] | 4 [3–113] | 1 [1–1] | 8 [1–17] | 2 [1–6] | 5 [4–6] | 1 [0–1] | 0 | 8 [6–9] | 11 [10–18] |
| **Mechanism of injury (%)** | | | | | | | | | | |
| MVA | 49 (28) | 1 (20) | 9 (27) | 2 (50) | 12 (31) | 0 | 4 (40) | 1 (100) | 1 (13) | 1 (33) |
| Motorbike | 45 (26) | 0 | 9 (27) | 2 (50) | 15 (39) | 1 (50) | 4 (40) | 0 | 1 (13) | 0 |
| Bicycle | 12 (7) | 1 (20) | 2 (6) | 0 | 0 | 0 | 1 (10) | 0 | 0 | 0 |
| Pedestrian | 60 (35) | 3 (60) | 11 (32) | 0 | 11 (28) | 1 (50) | 1 (10) | 0 | 6 (74) | 2 (67) |
| Other | 6 (4) | 0 | 3 (9) | 0 | 1 (3) | 0 | 0 | 0 | 0 | 0 |
| **Severity of head injury** | | | | | | | | | | |
| Head AIS | 5 [4–5] | 5 [5–5] | 0 [0–4] | 5 [5–5] | 3 [0–4] | 3 [3–3] | 3 [3–5] | 0 | 1 [0–3] | 0 [1–0] |

Data are expressed as median [quartile 1, 3], or n (%).

**AIS:** abbreviated injury scale, **CNS:** central nervous system, **HS:** Hemorrhagic shock/ exsanguination, **ICU:** intensive care unit, **LOS:** length of stay, **MOF:** multiple organ failure, **MVA:** motor vehicle accident, **Direct:** direct transportation, **ST:** secondary transfer, **TCA:** traumatic cardiac arrest.

278 in-hospital deaths recorded at 30-day (6%) in the trauma centers.

severity (AIS $\geq$ 3) which was negatively associated with secondary transfer (OR 0.1, 95% CI [0.02–0.3]). The Hosmer and Lemeshow test showed a good calibration of the model (p = 0.69). The discrimination as evaluated by the AUC was 0.76 (95% CI [0.73–0.78]) and the optimism adjusted AUC was evaluated at 0.75 (95% CI [0.72–0.77]). In the sensitivity analysis, the values of the OR of all the identified predictors were marginally modified by prehospital physiologic variables (S2 File).

The univariate comparison of the patients deceased at 30-day versus survivors is presented in the Supporting Information (S1 and S2 Tables). While crude mortality was significantly lower when patients were secondarily transferred (2% (n = 9) vs 7% (n = 269), p < 0.001), there was no significant association between 30-day mortality and the type of transport (OR 0.80, 95% CI [0.30–1.91]) when adjusted on predefined potential confounders in the multivariate analysis.

In total, 3538 (80%) of the 4412 patients from the Traumabase could be matched to the RAAR files. Among the 278 patients who died during the first 30 days in the TC, 251 (90%) could be matched, leaving 696 patients to investigate (Fig 2). The research performed in the medical records of the 8 central dispatch centers of the Île de France region (SAMU) allowed to identify 43 (4.5% of all death) who died in peripheral non-specialized centers. The remaining patients died on-scene (n = 569, 60%) or in TC (n = 40, 4%, missing or unmatched with records from the Traumabase) and 44 (4.5%) could not be found in the SAMUs' records (illustrated in the Flow Chart Fig 2). Those latter were considered as dead outside the TCs. So, the population based *undertriage leading to death* was 0.15%, 95% CI [0.12–0.18], and the overtriage was estimated at 60%, 95% CI [59–61].

The comparison of mortality between direct transport and secondary transfer groups could not be performed because no adjustment was possible given the available information in the RAAR and SAMU records. Nevertheless, considering the observed mortality in patients

**Table 2. Prehospital demographic, physiologic and injury characteristics of patients according to the type of transport (# included in the multivariate analysis).**

| | Direct transportation (n = 4031) | Secondary transfer (n = 381) | p |
|---|---|---|---|
| **Demography and outcome** | | | |
| **Age (year)** | 37 (17) | 43 (20) | < 0.001 # |
| **Male (%)** | 3118 (78%) | 293 (77%) | 0.89 |
| **BMI (kg/m$^2$)** | 25 (6) | 25 (5) | 0.82 |
| **ASA** | | | < 0.001 # |
| 1 | 2888 (72%) | 248 (65%) | |
| 2 | 869 (22%) | 106 (28%) | |
| ≥ 3 | 90 (2%) | 18 (5%) | |
| Unknown | 184 (5%) | 9 (2%) | |
| **AC/AP therapy** | 146 (4%) | 40 (11%) | < 0.001 # |
| **Professional situation** | | | < 0.001 # |
| Working | 1936 (63%) | 193 (59%) | |
| Student | 420 (14%) | 44 (13%) | |
| No activity | 595 (19%) | 74 (23%) | |
| Other | 118 (4%) | 18 (6%) | |
| **Mechanism of injury (all blunt)** | | | < 0.001 # |
| MVA | 1343 (33%) | 98 (26%) | |
| Motorbike | 1753 (44%) | 147 (37%) | |
| Bicycle | 171 (4%) | 41 (11%) | |
| Pedestrian | 668 (17%) | 83 (22%) | |
| Other | 96 (2%) | 12 (3%) | |
| **Hospital on-call period** | 2818 (70%) | 268 (70%) | 0.75 |
| **Ejection** | 1319 (42%) | 96 (38%) | 0.23 |
| **Global assessment of speed** | 2630 (76%) | 187 (58%) | < 0.001 # |
| **Death in the same vehicle** | 105 (3%) | 2 (0.8%) | 0.04 |
| **Entrapment** | 572 (14%) | 26 (7%) | < 0.001 # |
| **Unstable pelvic trauma** | 195 (5%) | 21 (6%) | 0.47 |
| **Traumatic cardiac arrest** | 99 (36%) | 27 (0.7%) | < 0.001 # |
| **Sum of Vittel criteria ≥ 1** | 3999 (92%) | 317 (83%) | <0.01# |
| **MGAP < 23** | 817 (21%) | 33 (11%) | <0.001 |
| **Prehospital variables** | | | |
| **SAP min (mmHg)** | 115 (25) | 122 (23)† | < 0.001 |
| **HR max (beats/min)** | 94 (23) | 90 (19)† | 0.01 |
| **SpO$_2$ min (%)** | 98 [95–100] | 98 [96–100]† | 0.85 |
| **Glasgow Coma Scale** | 15 [14–15] | 15 [15–15] | < 0.001# |
| **Pre-admission intubation (%)** | 896 (22%) | 22 (6%) | <0.001 |

Qualitative variables expressed as n (%) and quantitative variables as mean (SD) or median [quartile1, 3] according to the distribution.

†: average missing data 22–26% (not missing at random) **SAP** (Systolic arterial blood pressure): 22.8% missing, **HR** (Heart rate): 22.6% missing, **SpO$_2$** (peripheral oxygen saturation): 25.7% missing.

**BMI:** body mass index, **ASA:** American society of anesthesiologists, **AC/AP therapy:** anticoagulant and/or antiplatelet therapy, **MVA:** motor vehicle accident, **min:** minimal, **max:** maximal.

**Vittel Criteria:** French triage algorithm gathering 26 criteria to transport in a TC [18].

**MGAP:** Mechanism, Glasgow coma scale, Age, and arterial blood Pressure (prehospital score predictive of mortality) [20].

**Table 3. Severity, clinical characteristics and hospital course of trauma patients according to the type of transport.**

| | Direct transportation (n = 4031) | Secondary transfer (n = 381) | p |
|---|---|---|---|
| **Severity of injuries** | | | |
| **SOFA (day 1)** | 1 [0–4] | 0 [0–2] | 0.02 |
| **SAPS II (day 1)** | 17 [10–31] | 15 [10–25] | 0.03 |
| **ISS** | 11 [5–22] | 14 [9–22] | < 0.001 |
| **Major trauma ISS ≥ 16** | 1557 (39%) | 147 (48%) | 0.001 |
| **Head and neck AIS ≥ 3** | 900 (22%) | 103 (27%) | 0.04 |
| **Face AIS ≥ 2** | 468 (12%) | 29 (8%) | 0.02 |
| **Thorax AIS ≥ 3** | 1148 (29%) | 120 (32%) | 0.23 |
| **Abdomen AIS ≥ 3** | 408 (10%) | 76 (20%) | < 0.001 |
| **Extremities pelvis AIS ≥ 3** | 1115 (29%) | 58 (15%) | < 0.001 |
| **At admission** | | | |
| **SAP (mmHg)** | 129 (27) | 128 (24) | 0.63 |
| **Hemoglobin (g/dL)** | 13.3 (2.1) | 13.0 (2.0) | 0.04 |
| **Lactate (mmol/L)** | 2.5 (2.2) | 2.1 (1.5) | 0.01 |
| **Prothrombin rate (%)** | 82 (18) | 83 (15) | 0.12 |
| **Surgery in the first 24h** | 1941 (48%) | 152 (40%) | 0.01 |
| **Specific characteristics** | | | |
| **Hemorrhagic shock** | 266 (7%) | 14 (4%) | 0.03 |
| **Severe head trauma** | 405 (10%) | 4 (1%) | < 0.001 |
| **Spine trauma** | 451 (11%) | 85 (22%) | < 0.001 |
| **Medullary injury** | 116 (3%) | 22 (6%) | 1 |
| **Evolution and outcome** | | | |
| **Infection** | 176 (23%) | 16 (14%) | 0.03 |
| **ICU LOS (days)** | 2 [2–5] | 3 [2–7] | < 0.001 |
| **Hospital LOS (days)** | 7 [2–17] | 9 [5–16] | < 0.001 |
| **Discharge from ICU** | | | < 0.001 |
| Ward | 2938 (79%) | 326 (88%) | |
| Home | 697 (19%) | 34 (9%) | |
| Other ICU | 7 (0%) | 4 (1%) | |
| Rehabilitation | 82 (2%) | 7 (2%) | |
| **Day 30 mortality** | 269 (7%) | 9 (2%) | 0.001 |
| **Mortality over hospital stay** | 289 (7%) | 9 (2%) | 0.001 |
| **Predicted mortality by TRISS (%)** | 8.8% | 4.7% | 0.001 |

Qualitative variables expressed as n (%) and quantitative variables as mean (SD) or median [quartile1, 3] according to the distribution.

**SOFA**: sequential organ failure assessment, **SAPS II**: simplified acute physiologic score, **ISS**: injury severity score, **AIS**: abbreviated injury scale, **SAP**: systolic arterial blood pressure, **ICU**: intensive care unit, **LOS**: length of stay, **TRISS**: trauma related injury severity score.

**Hemorrhagic shock**: receiving ≥ 4 packed red blood cells concentrate within 6 hours.

**Severe head trauma**: Glasgow coma scale ≤ 8 and head AIS > 1.

directly admitted to the TCs (7.6%, 95%CI [6.8–8.4]), the range of systemic undertriage calculated was 1.5–2.8%, 95%CI [1.4–2.9].

## Discussion

The results of this study suggest that in the most densely populated French region, and within an exclusive trauma system, secondarily transferred patients were not at increased risk of

**Table 4. Results of the multivariate analysis to identify variables associated to the secondary transfer.**

| Variable | OR, CI 95% | p | Overal test F |
|---|---|---|---|
| **Intercept** | | <0.001 | |
| **Age** | | | 0.01 |
| [0,17] | 1.5 [0.9–2.3] | 0.07 | |
| ]17,44] | 1 | - | |
| ]44,64] | 1.4 [1.1–1.9] | 0.01 | |
| ]64, 100] | 1.9 [1.3–2.8] | 0.01 | |
| **Mecanism** | | | 0.01 |
| MVA | 1 | - | |
| Bicycle | 2.2 [1.4–3.4] | <0.001 | |
| Pedestrian | 1.5 [1.0–2.1] | 0.03 | |
| Motorcycle | 1.1 [0.8–1.5] | 0.44 | |
| Other | 2.0 [1.0–4.0] | 0.04 | |
| **Entrapment** | 0.6 [0.4–0.9] | 0.03 | |
| **Speed** | 0.6 [0.5–0.8] | <0.001 | |
| **GCS initial**[*] | 1.4 [1.3–1.5] | <0.001 | |
| **AC/AP therapy** | 2.0 [1.3–3.1] | 0.01 | |
| **AIS head $\geq$ 3** | 3.2 [2.4–4.3] | <0.001 | |
| **AIS face**[*] | 0.7 [0.6–0.9] | <0.001 | |
| **AIS abdomen $\geq$ 3** | 3.6 [2.6–4.9] | <0.001 | |
| **AIS pelvis $\geq$ 3** | 0.5 [0.3–0.6] | <0.001 | |
| **AIS head $\geq$3 * AIS abdomen $\geq$ 3** | 0.1 [0.02–0.3] | <0.001 | |

**OR, CI 95%:** Odds ratio and its 95% confidence interval.

**MVA**: motor vehicle accident, **GCS**: Glasgow coma scale, **AC/AP therapy**: anticoagulant or antiplatelet therapy, **AIS**: Abbreviated Injury Scale.

**Severe head trauma**: head AIS $\geq$ 3.

**Severe abdominal trauma**:

[*] Continuous variables

Akaike criteria: 2290

Hosmer Lemeshow: p = 0.70

AUC Roc curve: 0.76 [0.73–0.78]

Optimism: 0.0095

mortality after adjusting for confounding factors. This is consistent with the results of two meta-analyses which recently tried to evaluate the effect of secondary transfer on outcome and found no additional risk on mortality (pooled OR = 1.06, 95% CI [0.90–1.25][13,14].

## The trauma system performance

Our results can be partly explained by the structure of the trauma system in France which is entirely covered by a physician-staffed prehospital emergency care. This prehospital management was found associated with a significantly reduced 30-day mortality compared to paramedics in a study gathering 2700 patients with severe blunt trauma [15]. At the level of the TC, the observed rate of secondary transfer (9%) was lower than described in existing studies (ranging between 28% and 37% [32–35]). This can be explained both by a better initial orientation and by the available supply of TCs within the Île de France region. In the study by Garwe et al. (30% of secondary transfer), Oklahoma state was endowed of only one level 1 TC (and two level 2 TCs) for an average of 3.5 million inhabitants over 181.195 km$^2$ [33]. In the Île de

France region, one TC covered for an average of 2 million inhabitants over an average of 2.000 km$^2$. This ratio of TC to area cover might have allowed to cope with a higher overtriage (60%) than previously observed, and to maintain a low rate of undertriage.

The observed rate of secondary transfer remained higher than the recommended rate of "below 5%" [36]. Nevertheless, the patients that benefit most of early time-critical care, i.e. hemorrhagic shock or severe head trauma, were directly transported in 95% and 99% of cases respectively [37]. Most of the time, those patients are identified and triaged in the prehospital setting, according to the physiologic variables (for hemorrhagic shock) and to the GCS (for severe head trauma), except when their clinical status deteriorates secondarily.

### Determinants of secondary transfer

The main challenge in interpreting the determinants of secondary transfer is the discrimination between three types of cases: 1/ cases where initial severity was under-appreciated (undertriage) [33], 2/ cases that benefitted from initial stabilization in non-TC hospitals before transfer [12], and 3/ occasions of system saturation that triggered the initial work up to be completed in a non-TC hospital before confirmation as major trauma case and secondary transfer to a TC. These three situations could not be elucidated with the available data in our study. Despite this fact, the variables found to be associated with secondary transfer and the direction of these associations were in agreement with previously published studies [11,38].

Paradoxically, road users considered as more vulnerable, either by lack of physical protection such as pedestrians or cyclists, or due to age-related frailty, were more likely to be secondarily transferred. The challenge is actually the clinical recognition of traumatic brain injury, especially in elderly patients and/or after low energy transfer mechanisms (pedestrian or bicycle) [32,39]. Indeed, high-force injuries such as high energy transfer motor vehicle accidents are usually the prerogative of younger patients and also result in clinically more obvious injuries easier to triage, as exemplified with facial trauma, clinically obvious pelvic fracture or combined severe injuries which where protective factors of secondary transfer. In our study, secondarily transferred patients presented with higher injury scores (48% ISS $\geq$ 16 vs 39%) but had a lower non-adjusted mortality. This shows that, while considered as "undertriaged", these patients were properly cared for by the system, but obviously, their identification has to be improved in the region to streamline the triage process and to reduce the rate of secondary transfer.

### Population based analyses

The linkage of administrative databases to medical registries has blossomed in epidemiological studies in the last decade [40]. Trauma registries have historically provided the main source of data [41], targeting the most severely injured patients transported to a TC and focusing on hospital-centered outcomes. Linking to non-research database (e.g. administrative) now allows creating population-based injury databases, spanning all phases of care, gathering data on severe and non-severe patients. This is the first study in France enabling the drawing of the regional triage profile and the estimation of the death rate on scene within the studied trauma system. First, the rate of undertriage leading to death was low (0.15%, 95%CI [0.12–0.19]). Then, if we admit that the systemic undertriage was over 1.5%, the crude mortality in patients admitted to non-TC (secondarily transferred or not) was not significantly different than in directly admitted patients. And if we logically admit that patients would not die more when directly transported, then, the upper limit of our regional undertriage would be 2.8%, 95%CI [2.7–2.9]. Finally, death occurred on-scene in 60% of patients. This observation is in agreement with data from the US [42]. Many of these patients were too critically injured to survive

transport to any hospital, but some of these patients may be classified as preventable deaths. Indeed, the median time to death among patients with severe trauma is less than two hours [43]. The mean total pre-hospital time in our study was between 70 and 80 minutes, far longer than recently reported times in the PamPer (40 minutes) [44] or COMBAT trial (16 minutes) [45]. So, a more rapid transport to a TC might have prevented some deaths? To this concern, it is to note that in the physician staffed system in France, patients are carried in the mobile intensive care ambulance after immediate life-saving intervention, and resuscitation is continued on the way, even in futile cases. Thus, the observed 43 deaths in non-TC hospital might be a source of overestimation of the undertriage compared to other countries as the death is never pronounced during the transfer.

## Limitations

Our study presents some limitations. First, its retrospective design precluded the measurement and adjustment on confounding factors such as provider experience, quality, timing, or appropriateness of care at the initial facility which were not available in the databases. Second, multiple imputation was used to handle with missing data for most of the variables except for prehospital physiological variables for which data were not missing at random. Nevertheless, as the rate of missing data was low (max: 5.5%) for these variables, the potential resulting bias was likely to be limited. Third, a potential for selection bias (survivor bias) may have been introduced by severely injured patients deceased in local hospitals before transfer to TC. However, efforts have been made to evaluate it using the linkage of Traumabase and RAAR, and return to the archives from the 8 central dispatch centers of the region but adjustment could not be performed on these data as confounding variables were not available. A large observational study in France is about to begin and will allow to have an integrated database across all facilities of the participating regions to answer these points. An Australian study even showed that, in a well-resourced and coordinated trauma network, mortality was lower in non-TC when adjusted for age and injury severity [46].

Fourth, a probabilistic linkage had to be performed between the Traumabase and RAAR as both datasets did not have a unique common identifier and the performances of the matching algorithm we designed could not be evaluated before implementation. However, this method has already been used to match EMS records to hospital outcomes in the US [47,48] and has been validated among injured patients [49]. Fifth, the external validity of this kind of regional study is a complex issue but the present data shed light onto the triage process in an exclusive physican-staffed trauma system. Finally, beyond the part of uncertainty inherent to this type of epidemiological research, we intended to address the known potential pitfalls of such a process [40] and all computations were made to assess the maximum risk (including unidentified patients as dead in non-TC).

## Conclusion

Our study suggests that, in the context of exclusive trauma system with physician driven prehospital care, RTA trauma patients secondarily transferred to trauma center do not have an increased mortality when compared to directly transported patients. Patients identified as dead in non-specialized centers before transfer represented 4.5% of all fatalities.

## Supporting information

**S1 File. Database linking process.**
(DOCX)

**S2 File. Sensitivity analysis: Results of the multivariate analysis to identify variables associated to the secondary transfer with only "complete cases" for prehospital physiologic variables.** Complete cases included 3868 patients in the direct group and 262 in the secondary transfer), SAP remained in the final model but not HR and SpO2. As it can be seen by comparing to Table 4, the values of the OR of all the identified predictors were marginally modified in this sensitivity analyses. Data missing not at random are known to introduce bias, so we chose to withdraw these prehospital variables and we showed that it had no meaningful impact on our results with these sensitivity analyses.
(DOCX)

**S1 Table. Prehospital demographic, physiologic and injury characteristics of patients according to the 30-day mortality outcome.** BMI: body mass index, ASA: American society of anesthesiologists, AC/AP therapy: anticoagulant and/or antiplatelet therapy, MVA: motor vehicle accident, SAP: systolic arterial blood pressure, HR: Heart rate, SpO$_2$: peripheral oxygen saturation, min: minimal, max: maximal.
(DOCX)

**S2 Table. Severity, clinical characteristics and hospital course of trauma patients according to the 30-day mortality outcome.** SOFA: sequential organ failure assessment, SAPS II: simplified acute physiologic score, ISS: injury severity score, AIS: abbreviated injury scale, SAP: systolic arterial blood pressure, ICU: intensive care unit, LOS: length of stay. Hemorrhagic shock: receiving $\geq$ 4 packed red blood cells concentrate within 6 hours. Severe head trauma: Glasgow coma scale $\leq$ 8 and head AIS > 1.
(DOCX)

## Acknowledgments

To Jean-Louis Martin (Registre du Rhône) for his expertise in using RAAR files.

To Melanie D'Auria (from the French Ministry of the Interior) for her help in accessing timely to the RAAR database.

The Traumabase Group listed as contributors

LEAD AUTHOR FOR THE GROUP : **Anatole Harrois**, MD, PhD (Université Paris Sud and *Department of Anaesthesiology and Critical Care, Hôpital Bicêtre, Groupement Hôpitaux Universitaires Paris Sud, AP-HP, Kremlin Bicêtre, France*);

**Paer Selim Abback** MD, (*Université Denis Diderot and Beaujon University Hospital, Hôpitaux Universitaires Paris Nord-Val-De-Seine, Clichy, AP-HP, France*)

**Sylvain Ausset**, MD (*Anaesthesiology and Critical Care, Hôpital Interarmées Percy, Clamart, France*);

**Mathieu Boutonnet,** MD (*Anesthesiology and Critical Care, Hôpital Interarmées Percy, Clamart, France*);

**Thomas Geeraerts**, *MD, PhD (Department of Anesthesiology and Critical Care, Hospital, University Toulouse III Paul Sabatier, Toulouse, France)*;

**Anne Godier**, *MD, PhD (Université Paris Descartes and Department of Anaesthesiology and Critical Care, Hôpital Européen Georges Pompidou, APHP, Paris, France)*;

**Olivier Langeron**, MD, PhD (*Université Paris Est, Department of Anesthesiology and Critical Care, APHP, Hôpital Henri Mondor, 51 avenue du Marechal de Lattre de Tassigny 94010 Créteil, France)* ;

**Eric Meaudre**, (*Department of Anesthesiology and Critical Care, Military Teaching Hospital Sainte-Anne, Toulon, France, French Military Health Service Academy–Ecole du Val-de-Grâce, Paris, France)*;

**Catherine Paugam-Burtz**, *MD*, *PhD (Université Denis Diderot and Beaujon University Hospital*, *Hôpitaux Universitaires Paris Nord-Val-De-Seine*, *Clichy*, *AP-HP*, *France);*

**Romain Pirracchio**, *MD*, *PhD (Université Paris Descartes and Department of Anaesthesiology and Critical Care*, *Hôpital Européen Georges Pompidou*, *APHP*, *Paris*, *France);*

**Bruno Riou**, MD, PhD (Sorbonne Université, UMRS 1166, IHU ICAN; AP-HP *Department of Emergency* Medicine, *Groupe Hospitalier Pitié-Salpêtrière-Charles Foix*, *Paris*, *France*);

**Guillaume de St Maurice,** MD (*Anaesthesiology and Critical Care*, *Hôpital Interarmées Percy*, *Clamart*, *France*)**; Bernard Vigué**, MD (*Department of Anaesthesiology and Critical Care*, *Hôpital Bicêtre*, *Groupement Hôpitaux Universitaires Paris Sud*, *AP-HP*, *Kremlin Bicêtre*, *France*)

Traumabase préhospitalière

Killian Bertho, MD (Paris Fire Brigade, Emergency Medical Department, 1, Place Jules Renard, 75017 Paris, France.)

Charlotte Chollet-Xemard, MD (SAMU 94, Hôpitaux Universitaires Henri Mondor, APHP , Créteil)

Laurent GOIX, MD (SAMU 93, Hôpital Avicenne, Bobigny, APHP, France)

François-Xavier Laborne, MD (SAMU 91—SMUR de Corbeil, Centre Hospitalier Sud Francilien, 40, avenue Serge Dassault 91106 Corbeil-Essonnes Cedex, France)Paul-Georges Reuter, MD (SAMU 92, Hôpital Raymond Poincaré, Assistance Publique-Hôpitaux de Paris (APHP), Garches, Université Versailles-Saint-Quentin-en-Yvelines, Montigny-le-Bretonneux, France.)

David SAPIR, MD, (SAMU 77, Groupe Hospitalier Sud Ile-De-France, Hôpital de Melun 270 Av Marc Jacquet, Melun 77000 Cedex, France)

Benoit Vivien, MD (SAMU de Paris, Centre Hospitalier Universitaire Necker-Enfants Malades, Université Paris Descartes 149 rue de Sèvres, 75015 PARIS, France)

*The TRAUMABASE registry* provided all the data used. All the authors of the **TRAUMA-BASE** Group are responsible for the scientific content. The final version of the manuscript has been validated by the all the authors and the Traumabase Group.

## Author Contributions

**Conceptualization:** Sophie Rym Hamada, Jacques Duranteau.

**Data curation:** Sophie Rym Hamada, Nathalie Delhaye, Tobias Gauss, Mathieu Raux, Marie-Laure Devaud, Johan Amani, Fabrice Cook, Camille Hego, Alexandra Rouquette.

**Formal analysis:** Sophie Rym Hamada, Samuel Degoul, Jacques Duranteau.

**Investigation:** Marie-Laure Devaud, Johan Amani.

**Methodology:** Sophie Rym Hamada, Nathalie Delhaye, Samuel Degoul, Tobias Gauss, Mathieu Raux, Fabrice Cook, Camille Hego, Jacques Duranteau, Alexandra Rouquette.

**Project administration:** Sophie Rym Hamada.

**Resources:** Sophie Rym Hamada.

**Software:** Samuel Degoul.

**Supervision:** Alexandra Rouquette.

**Validation:** Sophie Rym Hamada, Alexandra Rouquette.

**Writing – original draft:** Sophie Rym Hamada, Nathalie Delhaye.

**Writing – review & editing:** Sophie Rym Hamada, Samuel Degoul, Tobias Gauss, Mathieu Raux, Marie-Laure Devaud, Johan Amani, Fabrice Cook, Camille Hego, Jacques Duranteau, Alexandra Rouquette.

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
