## [Decision Letter · Decision Letter 0]

2 Aug 2019

PONE-D-19-19735

Direct transport vs secondary transfer to level I trauma centers in a French exclusive trauma system: impact on mortality and determinants of triage on road-traffic victims

PLOS ONE

Dear Dr Hamada,

Thank you for submitting your manuscript to PLOS ONE. After careful consideration, we feel that it has merit but does not fully meet PLOS ONE’s publication criteria as it currently stands. Therefore, we invite you to submit a revised version of the manuscript that addresses the points raised during the review process.

We would appreciate receiving your revised manuscript by Sep 16 2019 11:59PM. To enhance the reproducibility of your results, we recommend that if applicable you deposit your laboratory protocols in protocols.io, where a protocol can be assigned its own identifier (DOI) such that it can be cited independently in the future. For instructions see: http://journals.plos.org/plosone/s/submission-guidelines#loc-laboratory-protocols

We look forward to receiving your revised manuscript.

Kind regards,

Zsolt J. Balogh, MD, PhD, FRACS

Academic Editor

PLOS ONE

Journal Requirements:

2. In the ethics statement in the manuscript and in the online submission form, please provide additional information about the database used in your retrospective study.

Specifically, please ensure that you have discussed whether all data were fully anonymized before you accessed them and/or whether the IRB or ethics committee waived the requirement for informed consent.

If patients provided informed written consent to have their data used in research, please include this information.

4. One of the noted authors is a group or consortium - The Traumabase Group.

In addition to naming the author group, please list the individual authors and affiliations within this group in the acknowledgments section of your manuscript. Please also indicate clearly a lead author for this group along with a contact email address.

Reviewers' comments:

Reviewer's Responses to Questions

**Comments to the Author**

1. Is the manuscript technically sound, and do the data support the conclusions?

Reviewer #1: Partly

Reviewer #2: Yes

2. Has the statistical analysis been performed appropriately and rigorously? 

Reviewer #1: I Don't Know

Reviewer #2: Yes

3. Have the authors made all data underlying the findings in their manuscript fully available?

Reviewer #1: No

Reviewer #2: No

4. Is the manuscript presented in an intelligible fashion and written in standard English?

Reviewer #1: Yes

Reviewer #2: Yes

5. Review Comments to the Author

Reviewer #1: needs a good statistician.

Not convinced by numbers.

otherwise a nice paper (maybe prehospital J might be more interested

Reviewer #2: Interesting study from a well developed trauma system in Paris.

My only major comment is that I don't get a sense of the total number of non trauma centres in that region, so the comment regarding "exclusive" trauma system is somewhat intriguing. Inclusive or exclusivity really depends on the proportion of major tertiary hospitals that are designated trauma centres (Level 1 or 2), so it may be argued that having 6 TCs in central Paris is really indicative of an inclusive system, depending on the number of similar sized tertiary hospitals within Paris.

The only other major issue is survivor bias for patients undergoing secondary transfer and the need to report matched mortality for patients not transferred to the TC. The authors have identified this I think, but some discussion on the need for an integrated database across all facilities, non trauma and trauma centres, is warranted given the volume of trauma being reported.

Lastly, I think a recent paper by Cornwall et al Injury 2019 on trauma transfers within a network in Australia outlining similar results would be worthwhile to place these results in context

6. PLOS authors have the option to publish the peer review history of their article (what does this mean?). If published, this will include your full peer review and any attached files.

Reviewer #1: No

Reviewer #2: No

---

## [Author Response · Author response to Decision Letter 0]

28 Aug 2019

All answers are provided in the Response. to reviewers file (10 pages)

---

## [Decision Letter · Decision Letter 1]

30 Sep 2019

Direct transport vs secondary transfer to level I trauma centers in a French exclusive trauma system: impact on mortality and determinants of triage on road-traffic victims

PONE-D-19-19735R1

Dear Dr. Hamada,

We are pleased to inform you that your manuscript has been judged scientifically suitable for publication and will be formally accepted for publication once it complies with all outstanding technical requirements.

With kind regards,

Zsolt J. Balogh, MD, PhD, FRACS

Academic Editor

PLOS ONE

Additional Editor Comments (optional):

Reviewers' comments:

Reviewer's Responses to Questions

**Comments to the Author**

1. If the authors have adequately addressed your comments raised in a previous round of review and you feel that this manuscript is now acceptable for publication, you may indicate that here to bypass the “Comments to the Author” section, enter your conflict of interest statement in the “Confidential to Editor” section, and submit your "Accept" recommendation.

Reviewer #1: (No Response)

Reviewer #2: All comments have been addressed

2. Is the manuscript technically sound, and do the data support the conclusions?

Reviewer #1: Yes

Reviewer #2: Yes

3. Has the statistical analysis been performed appropriately and rigorously? 

Reviewer #1: Yes

Reviewer #2: Yes

4. Have the authors made all data underlying the findings in their manuscript fully available?

Reviewer #1: Yes

Reviewer #2: Yes

5. Is the manuscript presented in an intelligible fashion and written in standard English?

Reviewer #1: Yes

Reviewer #2: Yes

6. Review Comments to the Author

Reviewer #1: All suggestions have been incorporated or discussed appropriately.

The paper is ready for publication

Reviewer #2: All my comments have been addressed and clarifying the structure of the trauma system in that region of Paris

7. PLOS authors have the option to publish the peer review history of their article (what does this mean?). If published, this will include your full peer review and any attached files.

Reviewer #1: No

Reviewer #2: No

---

## [Editor Report · Acceptance letter]

12 Nov 2019

PONE-D-19-19735R1 

Direct transport vs secondary transfer to level I trauma centers in a French exclusive trauma system: impact on mortality and determinants of triage on road-traffic victims 

Dear Dr. Hamada:

I am pleased to inform you that your manuscript has been deemed suitable for publication in PLOS ONE. Congratulations! Your manuscript is now with our production department. 

With kind regards,

on behalf of

Dr. Zsolt J. Balogh 

Academic Editor

PLOS ONE